# Monitoring and Assessing the Dynamics of Building Deformation Changes in Landslide Areas

**Lucyna Florkowska [1], Izabela Bryt-Nitarska [1,\*], Rafał Gawałkiewicz [2] and Janusz Kruczkowski [1]**

[1]   Strata Mechanics Research Institute, Polish Academy of Sciences, Reymonta 27, 30-059 Kraków, Poland; florkowska@img-pan.krakow.pl (L.F.); kruczkowski@img-pan.krakow.pl (J.K.)

[2]   Mine Areas Protection, Geoinformatics and Mine Surveying, AGH University of Science and Technology, alejaAdamaMickiewicza 30, 30-059 Kraków, Poland; gawalkie@agh.edu.pl

\*   Correspondence: nitarska@img-pan.krakow.pl; Tel.: +48-12-637-6200

**Abstract:** The paper presents the procedure and results of monitoring conducted by using a 3D measurement model, taking advantage of integrated surveying technologies developed for a building located within an activated landslide area. Geodynamic interactions within the building have resulted in a spatial deformation condition, leading to significant cracks of structure components and local basement floor upheavals. Conducted site research shows a reactivation of an old landslide form. To provide safe use conditions for the building, it was decided to monitor the structure and the area in its vicinity. Meeting this demand required developing an in-house monitoring system for the landslide form and the very structure. Measurements provided detailed information on the sizes and directions for the displacements of ground surface points and building structure points, as well as the dynamic properties of this phenomenon. Obtained results show the opportunity to use monitoring systems to acquire credible measurements data reflecting the real impact of ground landslide deformations on structures.

**Keywords:** landslide; mass movements; building diagnostics; damage; cracks; monitoring; headwater; slope

## 1. Introduction

The areas exposed to landslide movements pose a serious problem for providing development plans, due to a real risk to the infrastructure and structures [1–4]. In such areas, the basic tool for assessing the risk of landslides is the depth and surface monitoring of the directly affected area [5–9]. Monitoring should also cover the construction of building and engineering structures exposed to additional interactions from the landslide-affected ground [10–13].

The testing ground consisted of a landslide form and a public building, located in southern Poland, in the Wieliczka Foothills. This form qualifies for landslides on loess slopes [14–16]. Landslide processes intensified in May 2010 during heavy rain periods. Then, significant damage to the building structure caused by earth pressure was noticed. The progressive increase in damage was the basis for undertaking detailed diagnostic activities and developing a monitoring system. The aim of the actions was to determine the dynamics of landslide process and the method for handling the structure exposed to an emergency situation. Part of the diagnostic procedure was a 3D surveying model, making it possible to monitor the changes in landslide form and the structure construction that comprised of:

−   Measuring the landslide form by using absolute surveying coordinates—it confirmed the occurrence of slow mass movements in the ground surrounding the building; the maximum horizontal point displacement was 13.7 mm per quarter in the winter–spring period;

−　Measuring the observation (reference) points on the floor and building structure components in a relative system for the selected grid of control points that confirmed a spatial deformation of the building load-bearing system; the maximum horizontal displacement of floor points was 13.3 mm per quarter in the winter–spring period;

−　Measuring the deformations of building corners on a concentrated geodetic position (point) grid used to develop the calculation model to determine linear and angular deformations in building corner wall planes [17].

This paper makes references to the global measurements of landslide form and structures. The purpose of the work is to present the possibilities of a hybrid measurement model, combining various surveying techniques to observe changes that occur in a building located on a landslide slope. In addition, the article presents the consequences of the incorrect foundation of the structure and its exposure to the impact of geodynamic movement of the earth masses.

## 2. Materials and Methods

### 2.1. Geological Settings

Changes observed in the building and its vicinity made a basis for creating a landslide registration card number 12 19 055 (TZ 259) in March 2017 [18]. This card classifies the landslide as active, with variable intensity and the possible future enlargement of its surface area. It was found that the displacement of colluvial material occurs not only within the near-surface grounds, but affects also the deeper flysch layers. It was also found that the landslide covers the surface area of approximately 7.4 hectares, which is 152 m wide and 425 m long. The landslide slope is mildly inclined—approximately 8°—with a partly terrace-like course (Figure 1). Location of building complex within the TZ 259 landslide area and characteristic locations of damage.

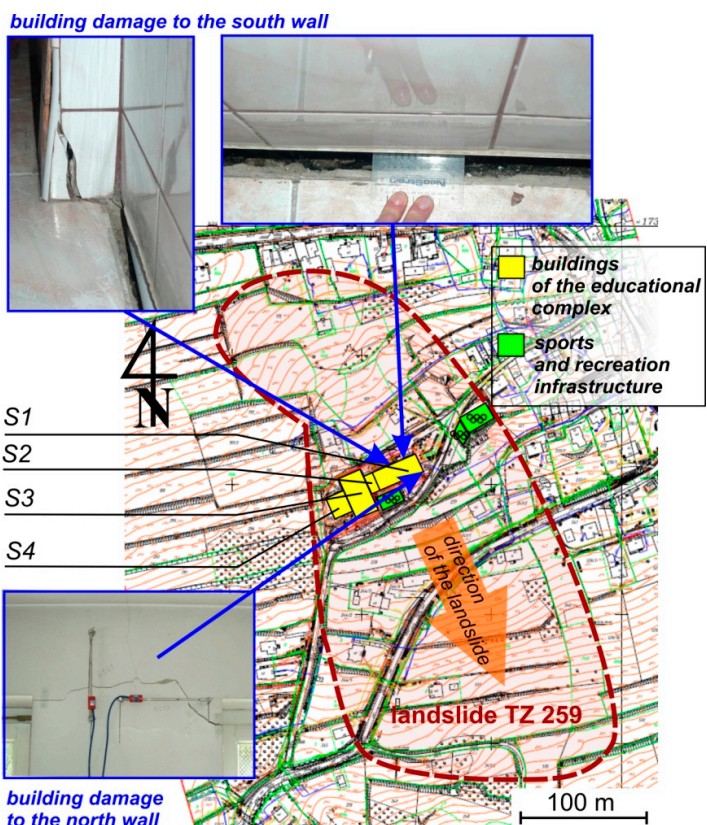

**Figure 1.** Location of building complex within the TZ 259 landslide area and characteristic locations of damage.

The slope is mainly occupied by an educational building complex. The body of the building complex has an extended projection with its longitudinal axis being almost perpendicular to the longitudinal landslide axis. The dimensions for complex horizontal projection are around 80.5 m × 33.0 m. The segment complex occupies more than one-half of the landslide range width.

The building complex consists of four segments: teaching building (marked S1) with dimensions of approximately 33.5 m × 15.5 m with a communication unit (marked S2) with dimensions of approximately 11.0 m × 15.0 m and sports and an entertainment arena (marked S3) with dimensions of approximately 33.0 m × 18.0 m with a community after-school club segment (marked S4) with dimensions of approximately 18.0 m × 18.0 m.

Detailed analyses resulted in finding that the landslide form could have been activated due to renewal, as a result of conjunction of three adverse events:

- Change in slope load conditions caused by extending the structure;
- Disturbance in natural groundwater drainage by locating an extended building complex across the slope and creating a drainage-free area, where accumulating water was absorbed by colluvial material layers;
- During long-term intense precipitation periods in 2010.

### 2.2. Measurement Model

The high-precision surveys conducted used a grid of geodetic points (positions) specially set up to implement monitoring. The dynamics of displacement processes in the landslide slope were determined by using surveys performed for nine months from 14 February 2017 to 24 November 2017. During observations, the area around the teaching complex and the very structure were monitored for displacements. Integrated surveying technologies were used for research purposes [19], including GNSS (Global Navigation Satellite Systems) surveys (reference points), polygonal grid (grid angular-linear measurements), and precise leveling.

The design of the control point grid was assumed, dividing the points into the following, as shown in Figure 2.

(a) Monofunctional i.e.,:

- Wall benchmarks were used to determine WH vertical displacements (57 points type 5F-75STK);
- 13RM21 Wall-mounted target plates used to determine horizontal WXY displacements (83 points).

(b) Bifunctional, in the form of earth points (in the ground—18 points) or floor points (in the floors—18 points), module-type Plastmark 50 PP-PLAS50 or survey steel nails with a center type 5F-75STK and 10TK-45, which were used to determine both WXY horizontal and WH vertical displacement values.

Measurements were performed by using:

- A GNSS set consisting of an R8s antenna and a Trimble TSC3 controller for the horizontal positioning and elevation positioning of grid references (static measurement at the reference point for 1 h 20 min) with relation to ASG-EUPOS system reference stations;
- TC 1800 Leica and 5503 DR Trimble total stations to determine the horizontal displacements of filling triangulation grid points;
- Precise Leica Na 3003 code digital level to determine the vertical displacements of points.

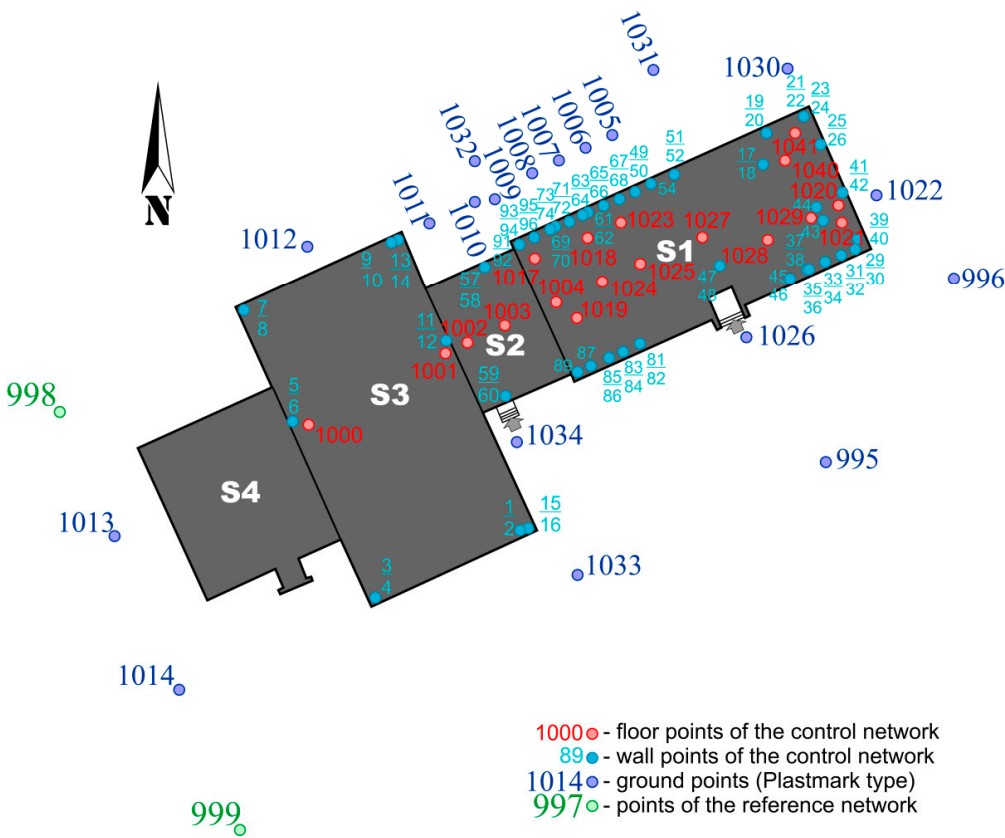

**Figure 2.** Schematic diagram of the location for horizontal geodetic grid reference points and the stabilized grid of control points around and inside the building.

The basic accuracy parameters of measurement sets are listed in Table 1.

**Table 1.** Basic accuracy parameters of measurement sets.

| Technology | Instrument | Measurement Accuracy | |
|---|---|---|---|
| | | **Horizontal** | **Vertical** |
| GNSS (static) | R8s Trimble antenna | ±5 mm + 0.5 ppm | ±5 mm + 1 ppm |
| Polygonal grid | TC 1800 Leica | ±3 cc (angle) ±1 mm + 2 ppm (distance) | - |
| | 5503 DR Trimble | ±9 cc (angle) ±2 mm + 2 ppm (distance) ±3 mm + 2 ppm [1] (distance) | |
| Precise leveling | Na 3003 Leica | - | ±0.4 mm/km (double levelling) |

[1] Measurement to target plate.

The site research was conducted and the condition of site class III geodetic state grid points was checked; the third elevation component was also determined, which allowed us to specify the suitability of those points for control works in progress, being eventually a component of the school monitoring system. The location of those points being disadvantageous in relation to the topographic details, significant distances to the object under analysis, and high calibration errors in control measurement using the GPS-RTN ranging from ±0.004 to ±0.213 m (flat 2000/21 system) and in a vertical plane at a considerable error scattering ranging from −0.122 to +0.065 m (Kronsztadt '86 system) disqualified the points as components of the control point reference grid. This resulted in the need for setting up our own reference grid around the building under examination, which would provide a basis for observations conducted inside the building (measuring target plates, making it possible to interpret horizontal displacements and a grid of benchmarks generating the information on vertical displacements).

Due to the nature of this paper and the need to obtain the most accurate positioning possible for the set control point grid located around the building complex and inside it, the geomorphological nature of slope under analysis, as well as low accuracy of the state geodetic grid point coordinates, it was necessary to establish a five-point reference geodetic grid. The stabilization works were carried out in February 2017. As part of the scheduled control measurement, two full measurement cycles for base points have been completed so far (997–999) using the GNSS static method (i.e., on 14 February this year and on 15 May this year). It allowed us to specify the consistency for retaining grid points. In addition, the analysis of coordinates from measurements 0 and 1 has provided information on the possible earth mass movement in the building foundation zone.

The maximum average error for point position (identical for all points under consideration) for measurement 0 was $m_{Pmax} = \pm 1.4$ mm, while for measurement 1 (identical for all points), it was $m_{Pmax} = \pm 2.2$ mm.

The retaining point grid shaped in this way in February 2017 was the basis for further surveying, which was aimed at determining the coordinates of the remaining control points located around the school building and inside it. The GPS points (997–999) fulfilled the role of the situation and elevation references. The remaining points of the main polygonal traverse inside the building body (floor points) were marked permanently in February 2017 (before starting the basis measurement) ("0" measurement) by using measurement nails with 10TK-25 and 10TK-45 center types, while wall-mounted points were marked with target plates (survey target films) sized 40 × 40 mm, type 13RM21. The schematic diagram for the location of all the horizontal geodetic grid points is presented in Figure 2.

The horizontal control point grid comprises of:

- Reference grid points (stabilization: modular point Plastmark 50 PP-PLA S50, benchmark 5F-75STK with center, measurement nail 10TK-45 with center)—five pieces in total;
- Control grid points located around the building (stabilization: modular point Plastmark 50 PP-PLA S50, measurement nail 10TK-45 with center)—18 pieces in total;
- Control grid points located inside the building (stabilization in the floors: modular point Plastmark 50 PP-PLA S50, measurement nail 10TK-25 with center)—18 pieces in total;
- Points of control grid located inside the building (wall stabilization in the form of 13RM21 target plates, the so-called tape survey targets 40 × 40 mm)—83 pieces in total.

The calculated values of the average control grid flat coordinate position errors range from 0.0 to a maximum of 2.4 mm.

To complement the spatial analysis of the geometrical changes of the building body in time, the points of the elevation grid inside the building were stabilized by installing steel-galvanized 5F-75STK-type wall benchmarks mounted in wall openings with heavy duty Fischer brand FIS VL (vinylester resin based) injection mortar. The layout of those points is shown in Figure 3.

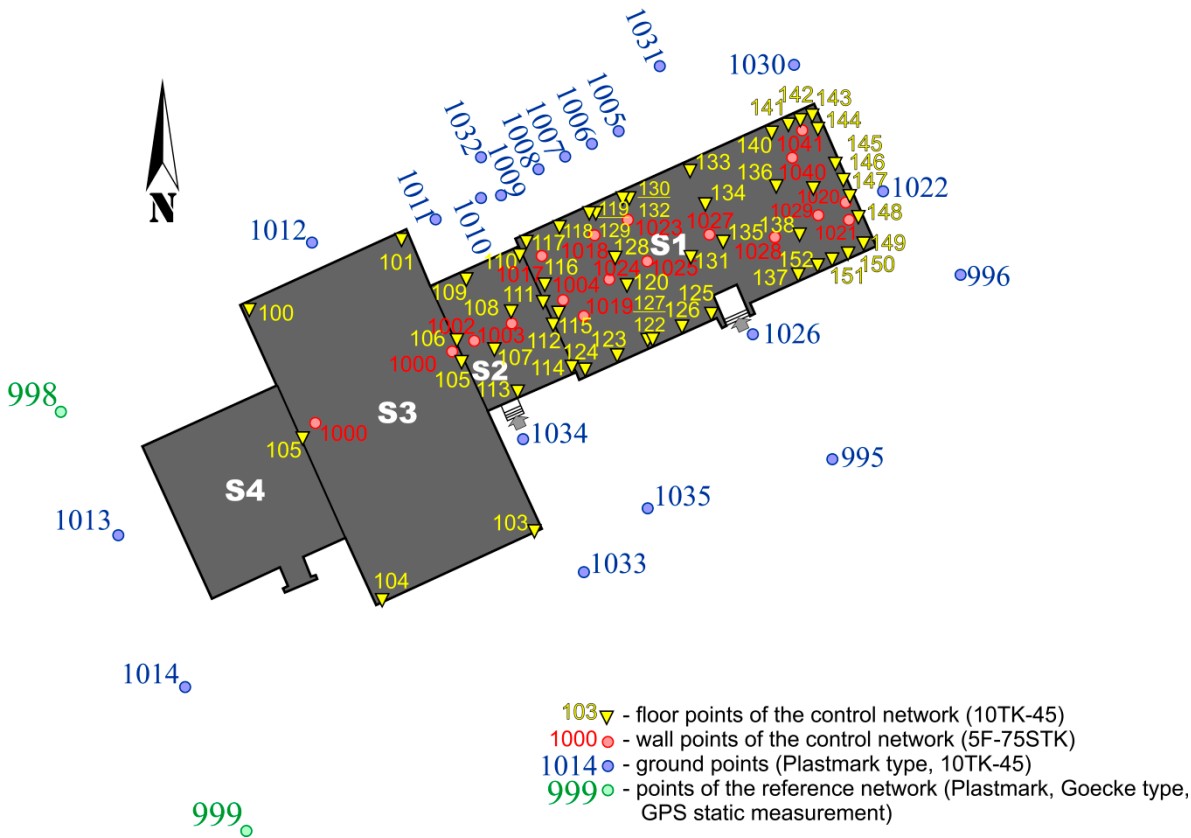

**Figure 3.** Schematic diagram of the location for elevation geodetic grid reference points and the stabilized grid of control points around and inside the building.

Due to mark geometry, the elevation grid of control points is composed mainly of horizontal geodetic grid points (bifunctional nature of marks), i.e.,:

- Reference grid points (stabilization: modular point Plastmark 50 PP-PLA S50, benchmark 5F-75STK with center, measurement nail 10TK-45 with center)—five pieces in total;
- Control grid points located around the building (stabilization: modular point Plastmark 50 PP-PLA S50, measurement nail 10TK-45 with center)—18 pieces in total;
- Control grid points located inside the building (stabilization in the floors: modular point Plastmark 50 PP-PLA S50, measurement nail 10TK-25 with center)—18 pieces in total.

In addition, the elevation mark grid was provided with control grid points located inside the building and stabilized with wall-mounted benchmarks type (5F-75STK)—52 pieces in total.

The average elevation error for benchmarks of the current measurement grid under analysis was $m_{Hi} = \pm 0.15$ mm.

### 3. Values and the Dynamics of Displacements

Surveys of displacements have shown that the area under examination is subject to slow-progressing, unstable displacement processes. Horizontal displacements are definitely dominating. This matches the type of movement as defined in the landslide card and specified as "landslip" [18]. The displacement of earth masses occurs generally in the direction of slope inclination.

It should be emphasized that the measurements taken define the surface (and building) movement characteristics. The analysis of deformation processes in deeper slope layers requires measuring with an inclinometer.

The dynamics of displacements was determined based on the magnitude of horizontal displacement vectors determined in individual measurement intervals (Figure 4).

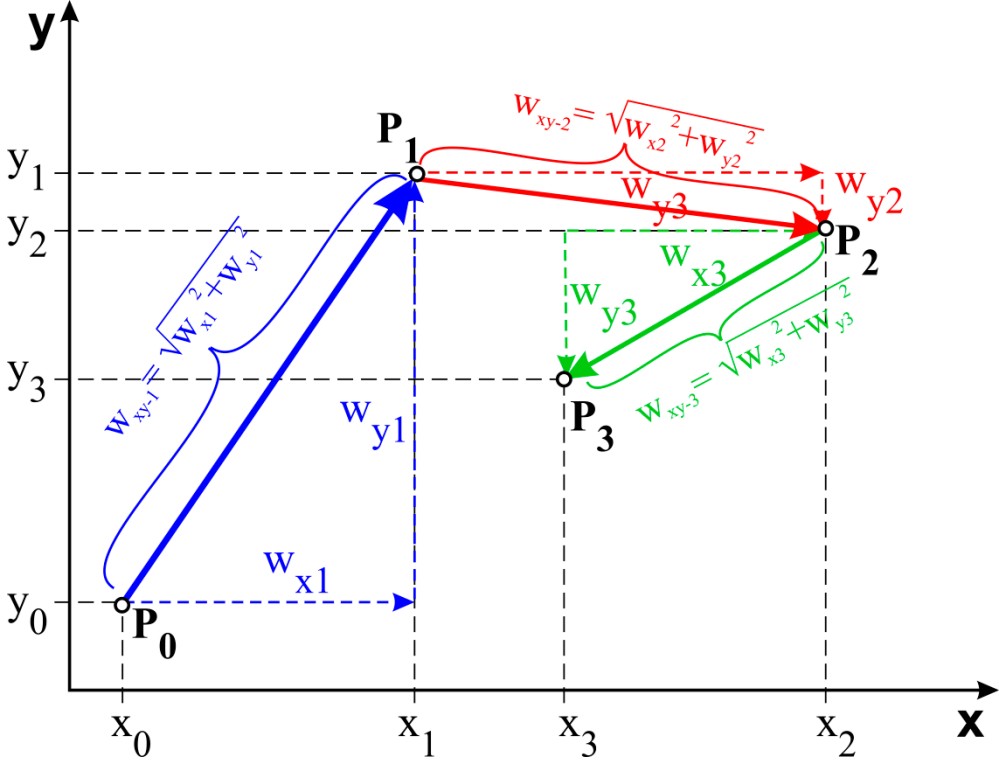

**Figure 4.** Magnitudes of horizontal displacement vectors.

Therefore, they show the values of horizontal displacement vectors (with no reference to vector direction and sense). The values of vertical displacements were not taken into account because of their negligible level (an order of magnitude smaller than that for horizontal displacements). The subsequent intervals were approximately three months long; the whole measurement period was nine months long: it took place between 15 February and 24 November 2017.

### 3.1. Land Surface Displacements

The highest changes were observed in the first measurement interval, i.e., between 15 February and 15 May 2017. During this period, the values of the terrain point horizontal displacements ranged from 6.0 to 15.5 mm. From among the earth geodetic grid (Plastmark type) points, the highest displacements showed 1034 and 1035 points located on the south side of the building, near the place where the communication unit (link) contacts the sports and entertainment arena. In general, the highest dynamics of changes should be linked with natural processes that occur at the end of winter and the beginning of spring when snow melts, the frozen soil thaws, and the ground becomes wetter while the groundwater flow intensifies.

Between 15 May and 21 August 2017, the displacement processes slowed down, which was probably due to the lower water saturation caused by the small total rainfall in the period.

Then, between 21 August and 21 November, the rate of changes increased again, especially in the points on the south side of the communication unit (link) and in points on the east side of the teaching building. At point 1026 in this period, the displacement process was the slowest. As a result of the destroyed measurement points (1010, 1011, 1009, 1008, 1007, 1006, 1005) located north of the building, it became impossible to continue measurements from May 2017 in this region. The dynamics of changes for horizontal displacement values is shown in Figure 5.

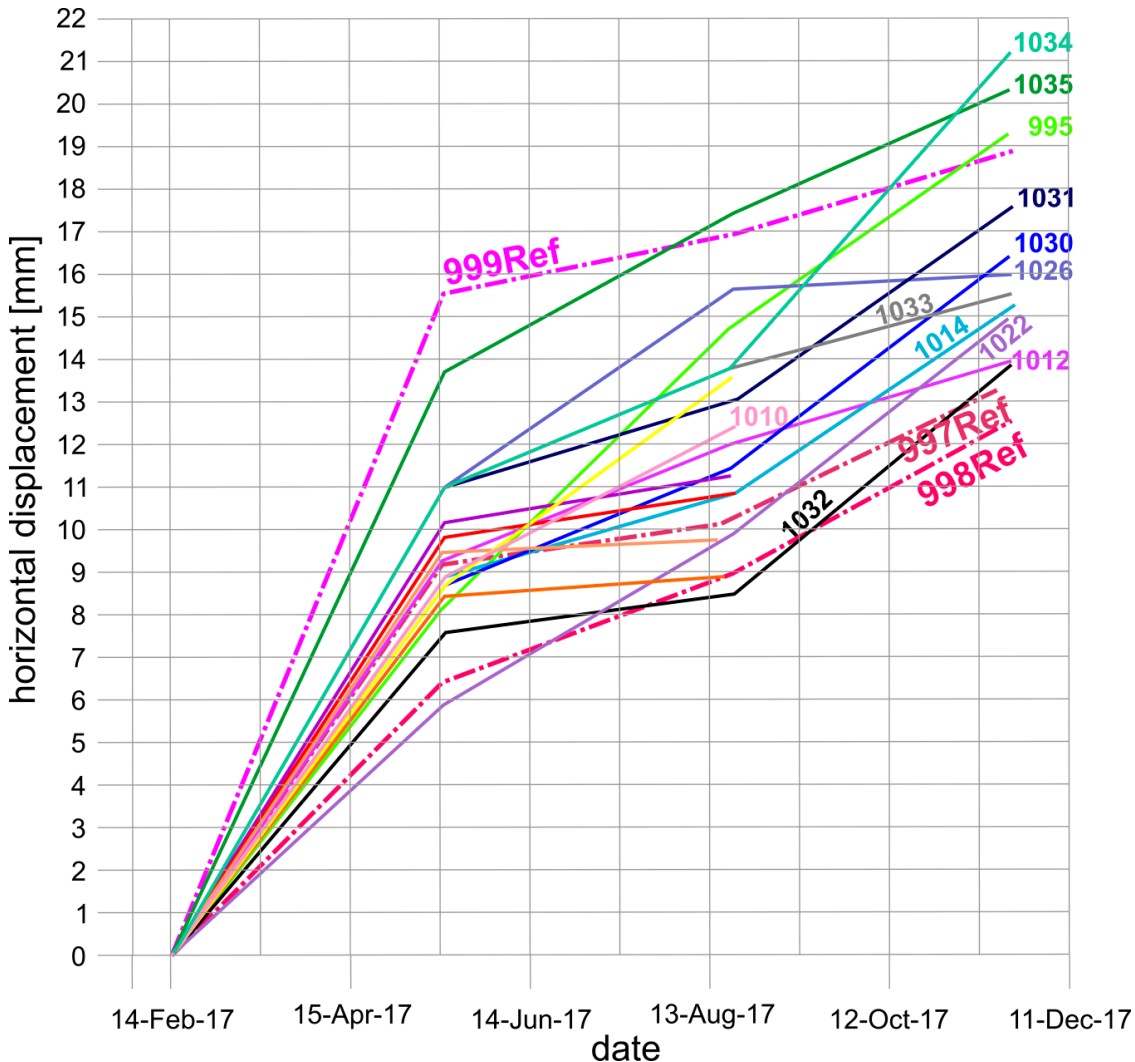

**Figure 5.** Dynamics of changes in values for horizontal measurements point displacements.

The image of movements comprised the measurements of vertical and horizontal displacements. Figure 6 shows the horizontal displacement vectors for control grid points observed from 14–15 February 2017 to 15–16 May 2017. The control point displacement values ranged from 7.6 mm to approximately 16.0 mm for points inside the structure and from 6.0 mm to approximately 13.7 mm for points around the building. In this period, it was possible to notice a clear trend to the translation of all the measurement points, along with a trend of the building complex to rotate in the direction of the slope inclination resultant. The map of horizontal displacements of controlled grid points showed a high variability in terms of the value and sense of vectors in individual building segments and in its closest vicinity, which is shown in the form of a hypsometric map in Figure 6.

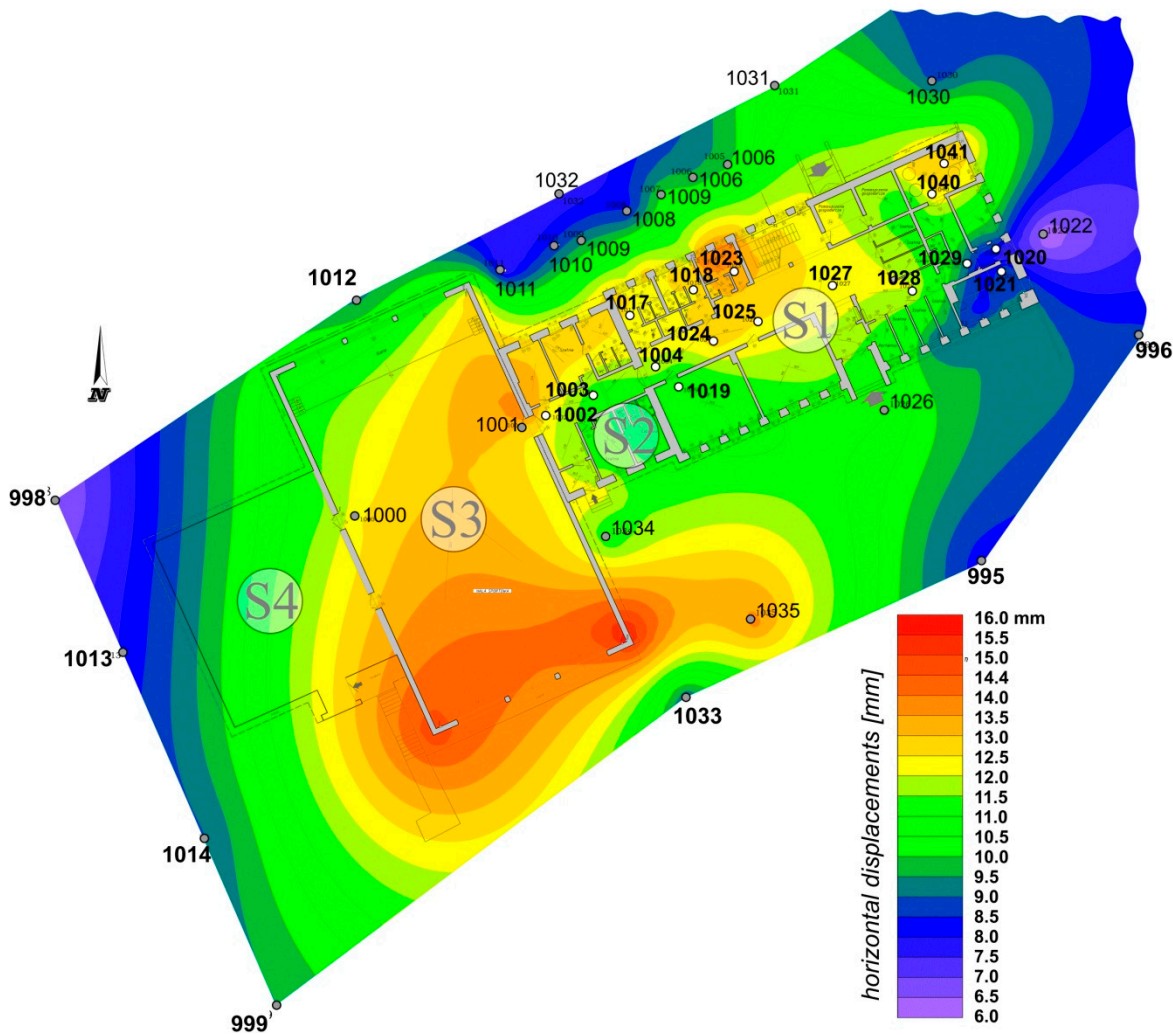

**Figure 6.** Hypsometric map for horizontal displacements of control grid points (period February 2017 to May 2017), the map uses flat coordinate system PL-2000/7.

By analyzing the monitoring results, it was found that the nature of ground displacement processes within the landslide under normal weather conditions is slow. The maximum rate of terrain point displacement measured in 2017 was:

– 13.7 mm per quarter in the winter–spring period (from 15 February to 15 May);
– 4.8 mm per quarter in the spring–summer period (from 15 May to 24 August);
– 7.3 mm per quarter in the summer–autumn period (from 24 August to 24 November).

On the other hand, the changes in the dynamics of displacement processes show a considerable sensitivity of landslide to seasonal and weather condition changes. There is a serious risk of accelerating the processes if extremely heavy precipitation occurs.

*3.2. Displacement of Buildings*

The dynamics of displacement processes that affects the teaching building (S1) is generally consistent with the dynamics of processes observed on the terrain surface. Changes in displacement increments in floor points (stabilized in the floor) in three measurement periods correspond to changes measured in terrain points (Figure 7). By analyzing the rate distributions on the horizontal projection, it can be noticed that in the eastern part of the building (hydrophore room, boiler room), the dynamic of displacement processes reaches its peak. There is a clear difference between the behavior at point

1041 (hydrophore room floor), where changes reach their peak values, and at point 1019 (near the communication unit), where the process rate decreased noticeably after high values were observed at the turn of winter and spring.

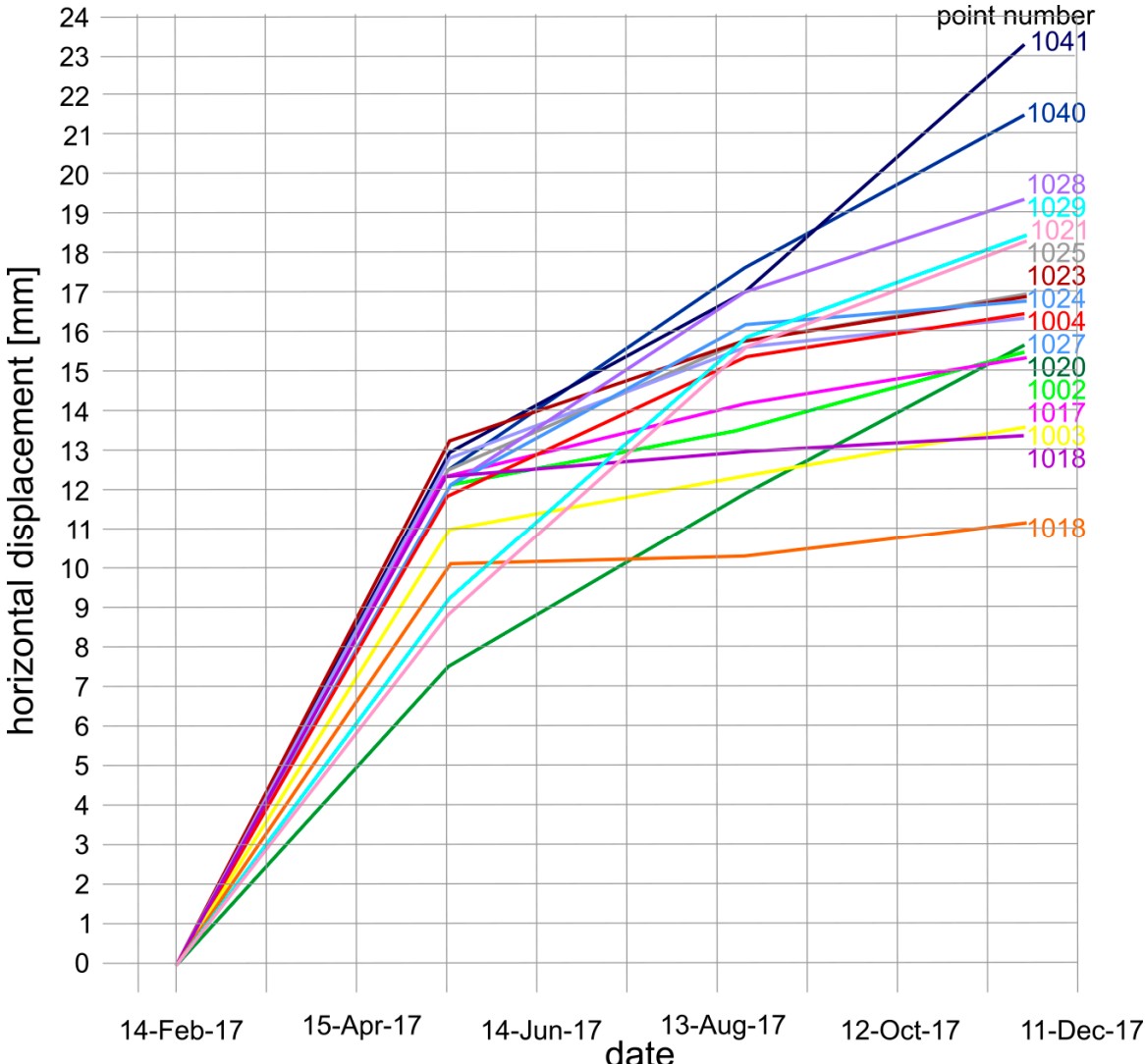

**Figure 7.** Dynamics of changes in values for horizontal measurement point displacements in S1 and S2 segment floors.

In segment (S3), in the first observation period, the horizontal displacement change dynamics in both floor points (1000 and 1001) were identical (Figure 8). Then, in the summer period of displacement slowdown, there was a difference in the behavior, which was expressed by the slower displacement of point 1001 from the side of the S2 communication unit than for point 1000, which was located on the opposite side of the hall. In the subsequent period (summer/autumn), a trend reversal was observed. In general, by analyzing measurements, the highest dynamics of changes were found in the winter–spring period, while a slowdown, not a stop interval, followed the peak period.

Comparing the dynamics for control point displacements distributed in the S3 sports and entertainment arena and in the S1 and S2 segments shows no re-acceleration of displacements in the sports and entertainment arena at the third observation interval.

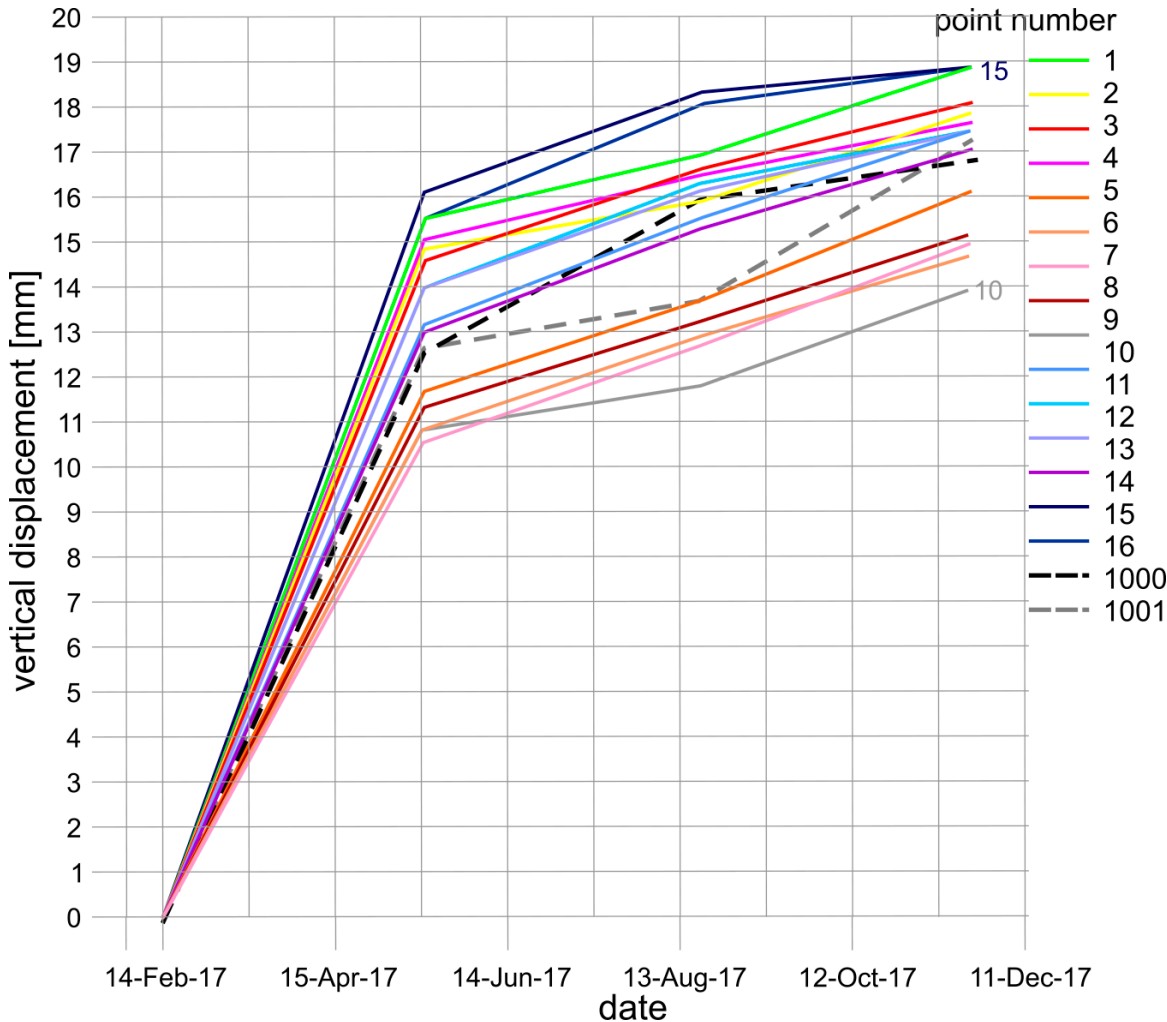

**Figure 8.** Dynamics of changes in values for horizontal wall and floor measurement point displacements in the S3 segment.

Extreme displacement values in three observation periods for all the measurement points are listed in Table 2.

**Table 2.** Horizontal displacements in individual observation periods.

| Measurement Place | Measurement Period | | | | | | Accuracy Min. [mm] |
|---|---|---|---|---|---|---|---|
| | 15 February–15 May | | 15 May–21 August | | 21 August–21 November | | |
| | Min. [mm] | Max. [mm] | Min. [mm] | Max. [mm] | Min. [mm] | Max. [mm] | |
| Terrain points | 6.0 (p. 1022) | 13.7 (p. 1035) | 0.9 (p. 1009) | 4.8 (p. 1033) | 0.4 (p. 1026) | 7.3 (1034) | |
| S1 segment floor | 7.6 (p. 1020) | 13.3 (p. 1023) | 0.2 (p. 1019) | 6.6 (p. 1021) | 0.7 (p. 1024) | 6.2 (p. 1041) | ±2.2 |
| S3 segment floor | 12.4 (p. 1000) | 12.6 (p. 1001) | 1.2 (p. 1001) | 3.6 (p. 1000) | 0.7 (p. 1000) | 3.4 (p. 1001) | |

The results of simultaneously conducted observations in the state of damage to building segments showed that there was an increase in the opening of the most significant wall fissures recorded from August 2016 to November 2017 in segment S1, with a maximum of approximately 4.7 mm (measured with a feeler gauge). It was also found that part of the fissures closed, which confirmed the spatial nature of building structure deformations resulting from the uneven slip of earth masses within the building complex foundations.

## 4. Summary and Conclusions

The paper presents a hybrid (integrated) measurement model, which was designed especially for the precise observation of buildings located on a landslide. As follows from the presented measurement results, this model fulfilled the role of both a scientific research tool as well as a system of detailed monitoring of changes in the state of the structure. The created system obtains a precise and complete view of changes occurring in the building and on the ground surface during landslide movement. Magnitudes and directions of displacements and their development over time were determined.

Therefore, it can be concluded that the created measuring system can be successfully used to study the displacements and observation of the condition of objects subjected to the impact of landslide movements, even with slow processes. The distribution of displacement values obtained on the basis of measurements can be used to estimate the state of strain and stress in structural elements.

**Author Contributions:** Conceptualization, L.F.; Methodology, R.G.; Validation, R.G., L.F., and I.B.-N.; Formal Analysis, L.F., I.B.-N., R.G., and J.K.; Data Curation, R.G.; Writing—Original Draft Preparation, I.B.-N.; Writing—Review and Editing, I.B.-N., L.F., J.K.; Visualization, L.F. and R.G. All authors have read and agreed to the published version of the manuscript.

**Funding:** This research was funded by Gmina Wieliczka and Strata Mechanics Research Institute of Polish Academy of Sciences.

**Conflicts of Interest:** The authors declare no conflict of interest.

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
