# Peer review of "Monitoring and Assessing the Dynamics of Building Deformation Changes in Landslide Areas"

_buildings, doi:10.3390/buildings10010003_

Round 1

Reviewer 1 Report

Consideration of the following comments and suggestions will improve the comprehension of the presented material.

Review comments

The word ‘condition’ from the title of the paper can be omitted. In the Introduction section lines 26-29 should be rewritten so that the 2010 events appear first and then the 2017 ones. The Introduction section should be expanded. The authors should clarify the purpose of this paper. The authors should write the location of the region they are working on. A photograph of the building and observed damages should be added. In Figure 1 the direction of the landslide should be shown with an arrow. A cross section (or schematic) showing underground layers of earth together with the building would be useful. Figure 4 should show the magnitudes of the horizontal displacement vectors. Can the authors mention the direction of the landslide in sentences 217-219? Was the direction for all three cases the same? In Figure 6 all the points mentioned in lines 229-243 should be highlighted because it is difficult to see them in the Figure. Sentence 262-264 should be split in simple sentences in order to understand what it is written. The conclusions are weak and not useful to other researchers. The author should generalize their results and give some guidelines how their research can be used.

Author Response

Thank you for all the comments submitted. Tips significantly improve the readability of the presented article. All suggestions were considered and taken into account. Thank you very much for your work and time. Yours sincerely, Authors

Reviewer 2 Report

The paper is well written, except for some minor grammatical errors. This study is relevant to the scope of the journal and the work is of very good quality. The manuscript is recommended for publication after addressing the following comments:

- Provide the reference in line 41 (page 1)

- Show the North direction in Figure 1

- Add average values to Table 2

- please provide an explanation for the locations of the maximum and minimum horizontal displacements shown in Figure 6.

Author Response

Thank you for all the comments submitted. Tips significantly improve the readability of the presented article. All suggestions were considered and taken into account.

We have left the existing solution in one area. Table 2 lacks average values because the directions of displacement of individual points are different. In such a situation, we decided that the average value, e.g. for all measuring points on the floor, is not reliable information. 

The reference in line 41 will be in the print version. At this point, the authors cited their own publication. 

Thank you very much for your work and time. Your sincerely, Authors.
